# Free PoC Testing for SARS-CoV-2 in Germany: Factors Expanding Access to Various Communities in a Medium-Sized City

**DOI:** 10.3390/ijerph19084721

**Published:** 2022-04-13

**Authors:** Anna Kristina Witte, Janina Grosch, Beate Conrady, Lena Schomakers, Marcus Grohmann

**Affiliations:** 1HTK Hygiene Technologie Kompetenzzentrum GmbH, Heinrichstr. 6, 96047 Bamberg, Germany; janina.grosch@hygiene-tk.de (J.G.); lena.schomakers@hygiene-tk.de (L.S.); marcus.grohmann@hygiene-tk.de (M.G.); 2Department of Veterinary and Animal Sciences, Faculty of Health and Medical Sciences, University of Copenhagen, 1870 Frederiksberg, Denmark; bcon@sund.ku.dk; 3Complexity Science Hub Vienna, 1080 Vienna, Austria

**Keywords:** SARS-CoV-2 point-of-care antigen tests, COVID-19 pandemic, testing strategy, testing behavior, Bürgertest, German citizen testing

## Abstract

During the third wave of the COVID-19 (coronavirus disease 2019) pandemic in Germany, free SARS-CoV-2 (severe acute respiratory syndrome coronavirus 2) point-of-care (PoC) antigen tests were offered to citizens at least once a week to prevent spreading by asymptomatic infected individuals. This study investigated user groups, timing, frequency, and test center locations in a typical medium-sized European city. We analyzed 27,369 pseudonymized datasets from eight centers over 12 weeks. Those were evaluated according to age, residence, appointment, and potential repeated test occurrence. The centers were visited by different groups; some centers were preferred by a predominantly younger demographic, whereas a mobile option attracted an older age group by reaching districts with few other testing possibilities. Elderly individuals were tested more spontaneously than younger individuals, and a test center at a ‘park and ride’ had more spontaneous visitors from outside of the city compared to other test locations. Only a small proportion of less than 4% came for testing more than five times. To preferably address many people for voluntary antigen testing, it is crucial to offer different test opportunities accounting for individual behavioral patterns, despite this requiring more complex and costly design than conventional forms.

## 1. Introduction

At the end of 2019, infections with SARS-CoV-2 (severe acute respiratory syndrome coronavirus 2) had been reported in Wuhan, China, for the first time. By the beginning of 2020, cases of SARS-CoV-2 had spread worldwide, and in March the WHO declared COVID-19 (coronavirus disease 2019)—the infectious disease caused by this virus—a pandemic [1]. So far, more than five million deaths have been reported in association with COVID-19 worldwide [2]. In Germany alone, the cumulative number of deaths with laboratory-confirmed COVID-19 exceeded 100,000 by the end of November 2021, demonstrating the relevance of the implementation of various measures [3].

The non-pharmaceutical interventions—including wearing masks, business closures, and contact reduction—were shown to be effective in the first two waves [4,5,6]. The three main measures used to mitigate the pandemic are lockdowns, vaccination, and SARS-CoV-2 testing. While vaccination has the purpose of increasing population immunity, testing is used independent of immunization to prevent infectious spread between persons by identification and subsequent isolation of infected patients—including asymptomatic infected persons—and quarantining of contacts. Since a few people with COVID-19 can account for a high number of secondary cases, removal of some of the most contagious persons could prevent case clustering [7]. The World Health Organization has implemented nucleic acid amplification test (NAAT) methods (e.g., real-time reverse-transcription polymerase chain reaction) as the reference diagnostic assay for SARS-CoV-2 detection [8]. Antigen-detecting tests can serve as a complementary method. These are less sensitive than NAAT, but feature rapid and less expensive detection of infections, especially if laboratory capacity is limited [9]. Different testing strategies have been suggested in the literature as measures to control the pandemic. These include weekly testing of all citizens [10,11], or ‘stratified periodic testing’, meaning that people with a higher risk of exposure to SARS-CoV-2 should be tested more regularly [12]. Potential limitations of frequent antigen testing programs must be considered, as on the one hand the lower sensitivity compared to PCR testing leads to a higher risk of leaving infectious people undetected, and on the other hand the relatively high number of false-positive tests can lead to unnecessary isolation. Another concern is the expense of broadly applied testing [13]. Addressing this, Gries and Welfins calculated that the costs of a random testing strategy, where a fraction of all age groups is regularly tested, are much lower than the negative economic and social impacts caused by lockdowns [14]. Additional compartmental epidemic modeling of even highly imperfect home-based rapid antigen testing indicates epidemiological and economic benefits of such nationwide programs [15]. A previous community testing pilot in Liverpool (UK) started in November 2020, and was associated with declining cases, but the precise impact alongside other parallel measures is under discussion [16,17,18].

The vaccination program in Germany started at the end of 2020; however, the number of vaccinated people increased slowly in the first months. Therefore, extended test possibilities were supposed to help to control the spread of SARS-CoV-2 and to limit the third wave in spring 2021 in Germany. Since asymptomatic infected people still can transmit the virus to others [19], each German resident without COVID-19 symptoms should have had the possibility of (at least) one SARS-CoV-2 point-of-care (PoC) antigen test per week (so called ‘Bürgertests’), free of charge. Negative test results were required to access several services, e.g., healthcare institutions, hairdressers, retail, or restaurants. For this purpose, several testing centers had to be established, many of which were affiliated with pharmacies. Furthermore, employers had to provide two SARS-CoV-2 tests per week for their employees.

Who actually used those free SARS-CoV-2 PoC antigen tests, and how often? Which needs can be addressed by different test modalities? Within the city of Bamberg (Bavaria, Germany), local authorities assigned a local institute to work out the capacity of tests available, independent of other regional private operators and pharmacies. Up to eight test centers with different opening hours and infrastructure were in operation during this period, with high infection and low vaccination rates, also supporting the local requirement of a negative test to enter restaurants or partake in cultural or tourist activities. We analyzed the first 12 weeks of testing when (1) the nationwide testing campaign in Germany began and (2) all eight test locations started operating consecutively. After this period, some of the test locations shut down. There is evidence of inequalities in large-scale community-based testing of populations regarding accessibility to test sites [18]. The main objective of this study was to evaluate details describing the use of test centers, and to connect these to individual factors of users, as an essential part setoff putting the cost and benefit of this public health measure into perspective, and to assist in minimizing inequalities. The location where the study was carried out has characteristic features of a medium-sized city in Europe [20], suggesting that the results can support a test strategy aiming to maximize the reach of people in comparable urban areas.

## 2. Materials and Methods

### 2.1. Test Center Locations

Eight test facilities in Bamberg city and rural district offered so called ‘Bürgertests’. Information about opening hours and days is provided in Table 1.

Locations were selected to ensure that every major city district provided an easily accessible test center, supported by a bus reaching areas without other test centers nearby.

### 2.2. Time Period of Analysis

A period of 12 weeks, from 27 March 2021 to 20 June 2021, was investigated, and represents the opening of all test centers operated by HTK Hygiene Technologie Kompetenzzentrum GmbH (Bamberg, Germany, assigned by the city of Bamberg). Within this time, all eight locations were open simultaneously (Table 2).

### 2.3. Use of Pseudonymized Data

To perform a PoC test, people could either book a time slot by using a booking app or could visit the test centers spontaneously if there was capacity. For the latter scenario, people could check in via QR code, or center employees could enter their data manually, either immediately or retrospectively. All people who were tested at all test centers agreed to the statistical analysis of their data with a privacy statement (either online when registered online, or by signing on paper when manually checked in). The software provider (KALA YOUR LIFE, Bamberg, Germany) offered data in pseudonymized form for data analysis. According to the provider, pseudonyms were generated using names, postal codes, and birth dates using the message-digest algorithm 5 (MD5) hash function [21]. For scheduled appointments, booking time, appointment time, and check-in time were recorded. Manually registered datasets were listed with check-in time only. Concerning test centers where appointments were not provided, all data were retrospectively inscribed by the testing staff. All retrospectively recorded datasets were thus excluded from analysis where the testing time was used.

During the first four weeks, different booking software (eTermin, Wallisellen, Switzerland) was used. Data were imported retrospectively to the new software by the software provider for subsequent pseudonymization and analysis. However, using the old system, only the time people booked the appointment was recorded, but not when they checked in. Thus, during the first four weeks, people who made an appointment but did not show up for testing were also analyzed (approximately 10%). 

Personal data of people registered with the online booking system and tested on 26th April were deleted due to incomplete data privacy statements on the first day of the new registration system. Only the time and number of the tests were recorded. Datasets without assignable postal codes, or with ages of less than 3 or more than 99, were excluded. 

### 2.4. Analysis

#### 2.4.1. Age

For each test location, the average and median ages of visitors were calculated, as the basis of calculation to determine the number of visitors to each test location.

#### 2.4.2. Region

By means of the indicated postal codes, visitors were classified to the urban district (‘city’, Bamberg), the rural district (‘district‘, Landkreis Bamberg), or neither. The numbers were analyzed for each location, and the percentages were compared to the total number of tests at each location.

#### 2.4.3. Scheduled versus Spontaneous Tests

All manually registered datasets were recorded as spontaneous. Furthermore, we categorized all online-registered tests as spontaneous if the time between registration and testing was less than two hours (including check-in via QR code). For each test location, the percentage was compared to the number of tests at each location or in the respective indicated time slot, weekday or day before holiday, age category, or resident of each location.

#### 2.4.4. People Tested Repeatedly

‘Returners’ were defined as visitors who came more than once for testing to one of the eight test locations. Those numbers were related to both the number of tests and the number of visitors (as indicated). As the basis of calculation for the percentage of joint visitors, the number of visitors to each test location was used. 

### 2.5. Metadata

Seven-day incidence (infection rate per 100,000 citizens) of the city and district of Bamberg was provided by the District Office Bamberg; information on vaccination data was provided by the Vaccination Center Bamberg and the government of Upper Franconia. Data about opening or closing steps were obtained from press releases by the city and the district of Bamberg, as well as the ‘Bayerischer Rundfunk‘ webpage (Bavarian broadcast) [22].

## 3. Results

### 3.1. Parallel Testing at up to Eight Locations over 12 Weeks

Extended testing of symptom-free people is one measure to control the COVID-19 pandemic. Thus, starting in mid-March, all German citizens in Germany were given the opportunity to have at least one free SARS-CoV-2 PoC antigen test (a so-called ‘Bürgertest’) per week. Since the infrastructure for such a large number of tests did not exist at this time, many pharmacies offered PoC testing, yet often not in sufficient quantities. In order to meet the required testing capacities, the city council of the urban district (hereafter called ‘city‘) of Bamberg (Bavaria, Germany), which includes 76,607 residents, along with 147,705 residents in its rural district (hereafter called ‘district’) [23], assigned a local institute. At the end of March, a central test center started operating at the central bus station (‘Central Bus Station’), and was extended to other locations to maximize the reach throughout the community. Test stations were provided in various locations in the city and district of Bamberg. In the district, one test center was located in a community center (‘District Community Center’). The test center locations in the city of Bamberg were a ‘park and ride’ car park (‘Park and Ride’), an office block (‘Office Block’), the theatre in the city center (‘Theatre’), a suburban property (‘Suburban Test Center’), and next to a tertiary care hospital (‘Hospital’). In addition, a bus (‘Bus’) was used as a mobile test center to accommodate nine different locations in the city where no or few other test options were available each week (Figure 1). An overview of the different opening periods and hours as well as average test numbers per (open) day is given in Table 1 and Table 2. After the here-described period of 12 weeks, the test centers at the theatre, as well as the mobile test center bus, were terminated. The suburban test center and the park-and-ride center followed one week later due to declining demand for tests, as demonstrated in Figure 2, where the decreasing 7-day incidence and increasing vaccination rate (in both Bamberg city and the rural district of Bamberg) are also depicted.

In total, 27,369 SARS-CoV-2 PoC antigen tests were carried out in this period, of which there were 3,645 datasets from eTermin, 18,066 regular datasets, and 5,658 manual datasets, of which 5,451 were retrospectively recorded. More than half of the tests were performed at the central bus station (15,981), which had the longest opening hours and most opening days. Tests were used by 18,449 different people, of whom most lived in the city and district of Bamberg (80%; more details in the following section).

### 3.2. Populations Visiting Test Centers Are Different

People tested in all test centers were on average 42 years (median: 40 years) old. Those attending the central bus station as well as the suburban test center were considerably younger, at an average of 38 and 39 years (median: 34 and 37 years), respectively, while the people tested on the bus (50 years average, 54 years median) and at the hospital (49 years average, 50 years median) were distinctly older (Figure 3A). 

As expected, most (80%) of the people tested were from the city (58%) and district (22%) of Bamberg (Figure 3B). However, we observed considerable differences between the locations. The district community center was mostly visited by people from the district (81%), while the bus had the highest proportion of citizens from the city, at 84%. In total, nearly all people tested on the bus (94%) were from the city or district of Bamberg. On the other hand, the lowest proportions of people coming from Bamberg city and local district were found at the park and ride (59%; including city (34%) and district (25%)) and the hospital (67%; including city (31%) and district (36%)).

The varying opening hours of the locations (Table 1) were intended to offer a wide range of testing options. A cross-comparison of whether people came at different times to the different locations is therefore impossible. 

### 3.3. Scheduled versus Spontaneous Testing

Whether people preferred to be tested in a scheduled appointment or spontaneously varied between locations. While a scheduled appointment was not offered at the bus at all, this option was technically possible but not known to the public for the hospital location resulting in exclusive and almost exclusive spontaneous usage at these locations, respectively. In the central bus station, the proportion of scheduled appointments was the highest, at 72%, while the lowest was at the park and ride, with 27% (Figure 4). Due to these different baseline scenarios, subsequent analysis was performed separately for each location. 

Concerning visiting times, it is worth mentioning that with 92%, the highest proportion of scheduled appointments was between 9:00 and 10:00 when the central bus station opened, which then decreased to 86% until the afternoon hours (13:00–14:00). On the four Fridays where the test location was open until 18:00, the percentage decreased towards the evening, down to 69%. Furthermore, at the office block more spontaneous tests were performed at noon (between 10:00 and 12:00 a.m.) than at other times (60% versus 73–78%) (Figure 5A and Appendix A). In most of the analyzed test centers, the older the individuals, the more spontaneous tests were registered (Figure 5B and Appendix A). Mostly, the scheduled versus spontaneous test ratio was similar on each weekday within each location. Exceptions were first the office block, where significantly more tests were performed through scheduled appointments on Fridays than on Tuesdays (75% versus 52%), and secondly at the community center, where most scheduled tests were on Fridays (70%), followed by Wednesdays (60%) and Sundays, where almost half of the tests were performed spontaneously (48%). Variations were also registered at the theater; however, there were also great differences in the numbers of performed tests due to variable opening hours (Figure 5C and Appendix A). The ratio between scheduled and spontaneous tests shifted to more scheduled appointments before public holidays in most locations—most notably in the office block. On days before a holiday, the ratio between scheduled and spontaneous tests increased to 82% more scheduled appointments, in comparison to only 67% on average at this location, and equivalent at the park and ride (36% versus 27%), theater (65% versus 54%), and suburban test center (62% versus 54%) (Figure 5D and Appendix A). However, no increase in test quantity was apparent on days before holidays (Appendix A). Lastly, there was a higher proportion of scheduled appointments from people resident in the city compared to residents of the district. This was most striking at the central bus station (city 74%, district 63%), the office block (72% versus 62%), the community center (located in the district, 64% versus 57%), and the park and ride (42% versus 32%) (Appendix A).

### 3.4. People Tested Repeatedly 

Pseudonyms of people were analyzed regarding whether and how often they reappeared (here called returners) at the same or a different location. In total, there were 1.5 tests per person on average (27,369 tests, 18,449 tested people), with an average age of 42 years (median: 40 years). Of all tests, 47% were used for people attending test locations more than once during the period under examination. In contrast, when analyzing only returners, there was an average of 41 years (median: 38 years) and 3.3 tests per person, visiting on 2.2 different weekdays and at 1.4 test locations. The highest share of tests from returners was recorded at the theatre and the central bus station, with 57% and 55% of the performed tests, respectively; the lowest percentages were found at the hospital (29%), the suburban test center (29%), and the park and ride (31%) (Figure 6). Altogether, a similar distribution of returners to one-off visitors was observed on weekdays, ranging between 44% (Monday) and 49% (Thursday). Visitors returning more than five times within the 12-week study period accounted for less than 4% of all tested people (629 from 18,449 tested people), and for 17% of all tests. Only 6% of these were tested consistently on the same weekday, and most were tested on three (27%) or four (29%) different weekdays.

Analyzing the occurrence of the pseudonyms at different locations revealed the highest number of matches for the central bus station with the theater (156 individuals visiting both locations), which accounted for 30% of all visitors at the theatre (corresponding to 2% of all visitors at the central bus station). Furthermore, the central bus station had quite a high number of matches with the office block (145, corresponding to 18% of visitors at the office block and 1% of the central bus station’s visitors); the lowest overlap was found between the theatre and the hospital, with only one person visiting both test locations (corresponding to 0.2% of visitors at the theatre location and 0.02% of the hospital’s visitors), and between the suburban test center and the hospital, which had four users in common (corresponding to 0.1% of visitors at the suburban test center and 1% of the hospital’s visitors) (Table 3).

## 4. Discussion

In addition to vaccination and reducing contact, increased testing is one of the major measures used to control the COVID-19 pandemic [24]. Since March 2021, each resident in Germany has been entitled to receive one free SARS-CoV-2 PoC antigen test per week; therefore, the necessary infrastructure had to be established quickly. The municipality of the Bavarian city of Bamberg assigned a local institute to supply the capacity required. While running up to eight locations in parallel, we aimed to analyze how the different no-cost testing sites were used, and the kind of audience that each testing modality attracted.

The data demonstrate the distinct requirements of different population groups. The bus as a mobile option was utilized mostly by the older age group living in the city. Additionally, spontaneous testing seemed to be preferred by older age groups, as demonstrated in most test centers. The fact that appointments could not be scheduled at the bus may have also been attractive to the older age group. Whether the online registration format or the process of scheduling an appointment itself was the reason remains an open question. The mobile character of the bus aimed to offer SARS-CoV-2 PoC antigen tests close to the individuals’ places of residence. Indeed, we can show that few visitors here were tested at other locations. The regional character was also demonstrated in the district community center, where mostly people from the district were tested. The highest share of returning visitors at the theatre test center may be attributed to a high number of theater subscriptions. The rather high exchange of returners between the central bus station and the theatre could be due the local proximity of both locations. The park and ride test location also showed noteworthy characteristics—the ratio of scheduled appointments was very low (27%), and the percentage of people not originating from Bamberg was rather high (41%). The rather low proportion of returners suggests that many spontaneous visitors utilized a SARS-CoV-2 PoC antigen test before entering the city. 

The age of people tested next to the hospital was the highest, averaging 49 years (median: 50 years). People using this test center were most likely either hospitalized or visiting someone at the hospital. For both scenarios, negative tests were required. Older people are more often hospitalized (in Germany, approximately half of the patients were more than 65 years old in 2019 [25]), and in the case of this specific hospital, the average age in this period (April/May/June) was 53 years, likely explaining the higher average age at the corresponding test center.

Altogether, our results were well explained by the test centers’ locations and organizational circumstances. This confirmed that the intention of creating diverse test locations aiming to address as many people as possible was achieved. A comparable testing approach was established in Worchester (USA) to offer low-barrier testing throughout the community [26]. Furthermore, our study confirms the preference of distinct populations for shorter distances to test centers, including the elderly population, as Hernandez et al. found during the investigation of COVID-19 testing services in New Orleans (USA) [27]. This indicates that these findings can be applied to various comparable urban areas. However, it is worth noting that the bus test center was rather laborious and expensive due to higher personnel costs and assembly at the stations. This additional outlay is also reflected by the lowest average number of tests per day. In case larger testing capacities are reinstated by the government, aiming to include as many people as possible, this higher expenditure must be considered.

As mentioned above, the option to schedule an appointment had various uses. Not only did the older population prefer to use spontaneous testing, but also the pattern for each location differed considerably—it ranged from 27% to 72% on average between the test centers for scheduled appointments. More scheduled appointments were recorded before holidays, which may have been caused by planned meetings with family and friends. Offering both options therefore seems to be the best solution to reach as many people as possible.

Controlling the pandemic with regular weekly tests of every citizen, as suggested by Peto [10,11], would result in almost 80,000 performed tests per week for the city of Bamberg. This was nowhere near reached according to our dataset. Other calculations for ‘universal random testing’ suggest testing 27% of the population daily [12] (corresponding to approximately 150,000 tests per week in Bamberg), which was even further out of reach. On average, the eight test centers reached utilization peaks of up to almost 5% if calculated with the inhabitants of the city (if residents of both the city and the district were accounted for, only 2% of the population was reached). The capacities were clearly not sufficient for reaching the test numbers of the two calculations mentioned above. Even though several more test providers were situated in the city of Bamberg, the presented data correspond to a major proportion of the testing capacity for the city (information from City Office Bamberg). A significant number of tests outside of our cohort therefore seems unlikely, meaning that the observed trends reflect the conditions. In addition, we noted that the availability of SARS-CoV-2 PoC antigen tests was still not exhausted, showing the discrepancy between theoretical requirements and actual use of testing. Consequently, testing capacities were not increased, and at the end of the investigated period (June 2021) locations even had to close due to further decreasing demand. Whether this was caused by decreasing infection rates in summer 2021 and associated opening steps with fewer requirements for negative SARS-CoV-2 tests, increasing numbers of vaccination rates, or warmer weather where people meet predominantly outside, remains hypothetical. Nevertheless, the analysis of Gabler et al. indicates that testing even 10% of the population weekly had a large effect on reducing new infections in spring 2021, using data from Germany [28]. No correlation was identified between the infection/vaccination rates and the number of tests (data not shown), which may be because of our investigation of a short period of 12 weeks, small sample size, or because testing behavior is very complex and can be influenced by various external factors. In conclusion, regular tests once a week on a voluntary basis are seldom performed. Our results of pseudonymized data showed that people getting tested more than five times in the 12-week period accounted for less than 4% of all tested people. This number might be slightly underestimated, since the software provider created different pseudonyms even when a typing error occurred. Additionally, people had the choice of testing at home, at work, or at external test centers. Even with a high probability of underestimation, our study shows that the offer of one test per week and restrictions on access to public places are not by themselves sufficient to accomplish regular testing of the population. This should be taken into consideration in further discussion about tests that are free of charge for all residents as a measure to control the pandemic. Further studies at later stages of the pandemic, over longer periods, and in other cities might be helpful to gain a better understanding of the best ways to improve testing rates. Interestingly, the number of tests with scheduled appointments was higher before holidays, suggesting that tests were more often performed before meetings with family or private activities, and not for reasons of interest or work. This is consistent with a survey we started in parallel, where people most frequently indicated meeting with friends and family as motivation for testing [29]. This suggests that events—regardless of mandatory tests for public or voluntary tests for private gatherings—are a better trigger for performing SARS-CoV-2 PoC antigen tests than personal interest or social responsibility. 

### Limitations of the Study

As mentioned throughout the discussion, there are limitations of this study, which can be summarized as follows: Due to typing errors, the number of pseudonyms might be overstated and, thus, the returning visitors underestimated. Furthermore, data from other test centers or how often people tested at home or at work were not part of the present study. The focus here was instead on whether and how often different test centers were visited. The dataset was not sufficient to find a correlation between infection rates and numbers of tests, leaving the most interesting question of the impact of frequent rapid testing in reducing SARS-CoV-2 transmission unanswered. Other earlier implementations of antigen testing at the population level modeled an additional effect on the reduction in COVID-19 cases, such as a study of three consecutive weekends of nationwide mass antigen testing in Slovakia [30], but dealing with the same limitations of different parallel control measures. Even a long-term dataset might not be able to precisely unravel the complex interactions, including different restrictions, holidays, weather, vaccination rates, virus variants, etc. 

## 5. Conclusions

Free-of-charge point-of-care SARS-CoV-2 testing started in Germany in the middle of the third COVID-19 wave. Vaccination has increased to approximately 70% (middle of December 2021), but failed to meet expectations at the beginning of the fourth wave. If more testing is required to control this or a future pandemic, more people must be reached. Although compulsory SARS-CoV-2 antigen testing at the workplace was introduced in autumn 2021, a higher variation of test modalities should be considered in order to maximize voluntary testing, as not everybody is included in compulsory testing. Only slightly more than 20% of all visitors returned for repeated testing during the 12 weeks monitored. Our results confirm the usage of easily accessible services and the importance of adjusted services for different user groups. Of particular note are stationary or mobile test centers in peri-urban areas, representing a low-barrier concept that can reach the older demographic. However, it must be considered that tailored solutions, such as a mobile bus as a test center, are considerably more expensive and time-consuming, and require a greater organizational effort compared to regular test sites. Our findings were obtained from a mid-sized urban area with common characteristics, and should be applicable to many areas.

## Figures and Tables

**Figure 1 ijerph-19-04721-f001:**
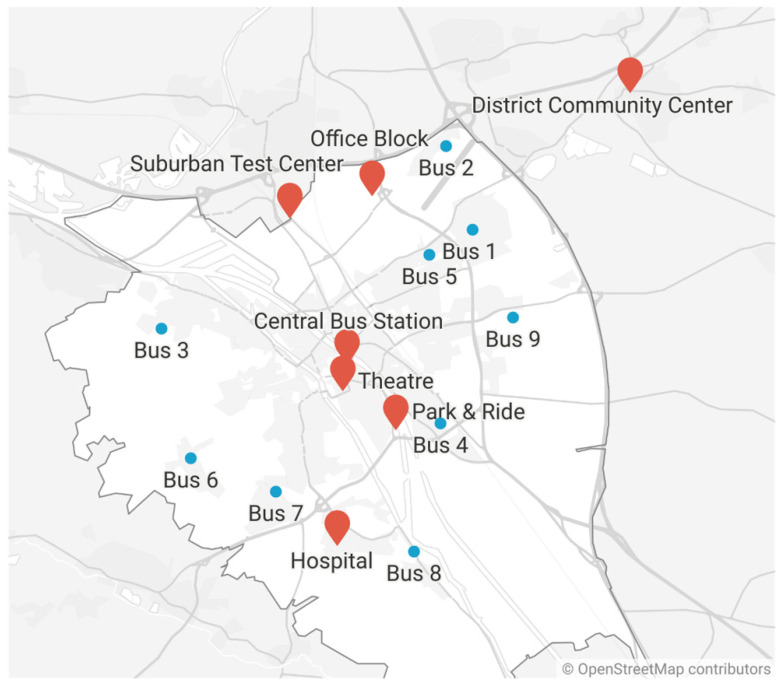
Test centers in Bamberg. Locations of the test centers are illustrated as red markers on the map of Bamberg. The different bus stops are depicted in blue (stops 1 and 2 on Tuesdays, stops 3 and 4 on Wednesdays, stops 5 and 6 (stop 6 started with calendar week 18) on Thursdays, stops 7 and 8 on Fridays, and stop 9 on Saturdays (stop 9 until calendar week 18 on Thursdays)). The city is highlighted in white, while the district is shown in grey. The graphic was created with Datawrapper (online tool provided by Datawrapper GmbH, Berlin, Germany, Version 1.25.0).

**Figure 2 ijerph-19-04721-f002:**
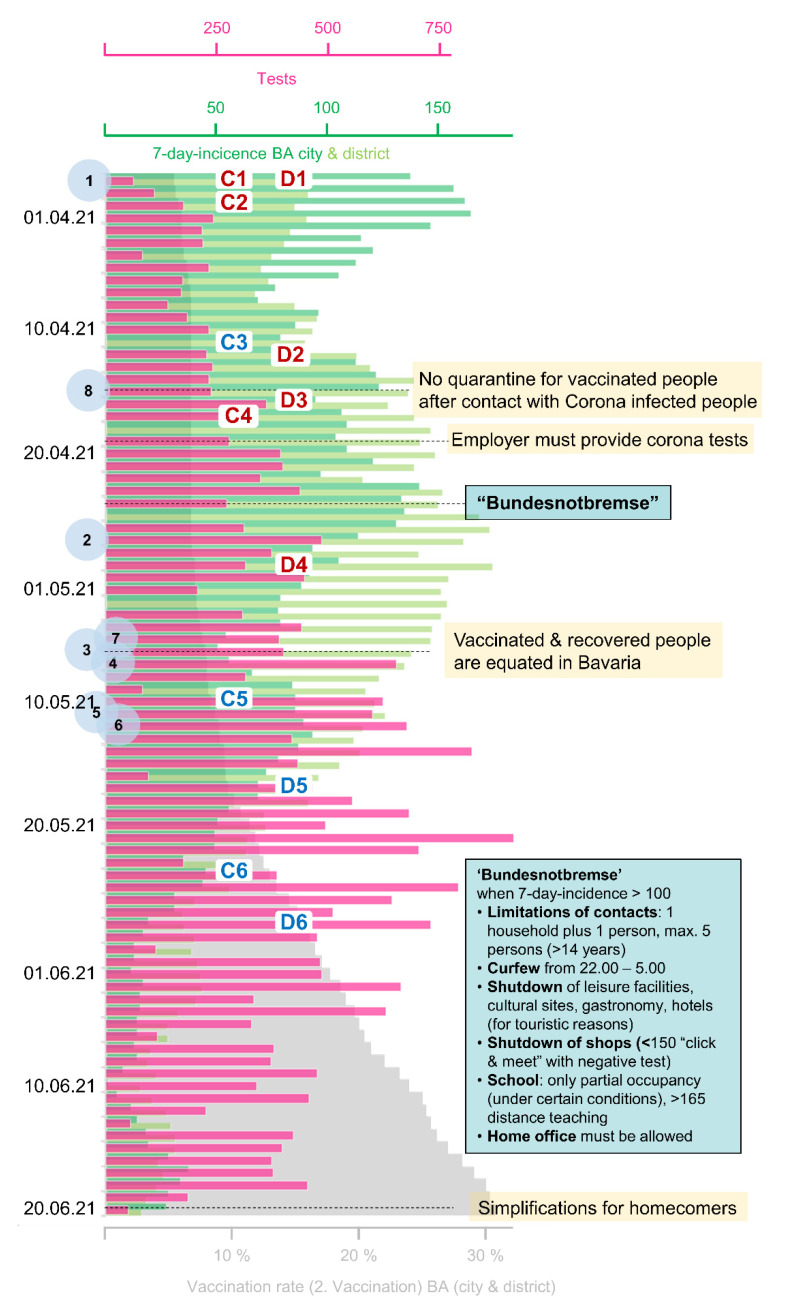
Overview of performed SARS-CoV-2 PoC antigen tests, 7-day-incidences, vaccination rates, and measures. The number of tests from all eight test locations is shown in pink. The blue circles indicate when the different locations started testing (1: central bus station, 2: office block, 3: theatre, 4: district community center, 5: park and ride, 6: suburban test center, 7: hospital, 8: bus.). The 7-day infection rates per 100,000 citizens are demonstrated separately for the city (dark green) and the district (light green) of Bamberg, and the vaccination rate is documented for the city and district together (grey). Important events concerning COVID-19 are noted on the right-hand side (yellow). Restrictions and easing of restrictions specific to Bamberg city (C1–C6) and district (D1–D6) are tagged in red (restriction) and blue (easing). The most severe measures of the ‘Bundesnotbremse’ (‘federal emergency brake’)—the section of law that regulated the measures for Germany during the pandemic for approximately two months (end of April until end of June 2021) at the national level—are summarized in the text box. In Bavaria, similar measures have been applied previously, dependent on the infection rates. Restrictions (red) and easings (blue) for the city (C) and district (D) of Bamberg (marked with numbers in the middle of the graph): After three days over or under a certain infection rate (7-day-incidence per 100,000 inhabitants, referred to as 7di), the measures were active and inactive, respectively, following a two-day transition period (date marked in the graph): **City:** C1. Nurseries closed; C2. (>100 7di): Contact limitations, curfew, shutdown (click and collect possible), cultural sites closed; C3. (<100 7di): Less strict contact limitation, no curfew, click and meet (with negative test), cultural sites with appointments; C4. (>100 7di): Contact limitation, curfew, click and meet (with negative test), cultural sites closed; C5. (<100 7di): Outdoor gastronomy can open; C6. (<50 7di): No tests necessary for gastronomy, culture etc. **Rural District:** D1. (>50 7di): Contact limitation, curfew, click and collect, cultural sites with appointments; D2. Nurseries closed, distance teaching; D3. (>100 7di): Stricter contact limitations, curfew, click and meet (with negative test); D4. (>150 7di): only click and collect possible; D5. (<100 7di): Outdoor gastronomy can open; D6. (<50 7di): No tests necessary for gastronomy, culture etc.

**Figure 3 ijerph-19-04721-f003:**
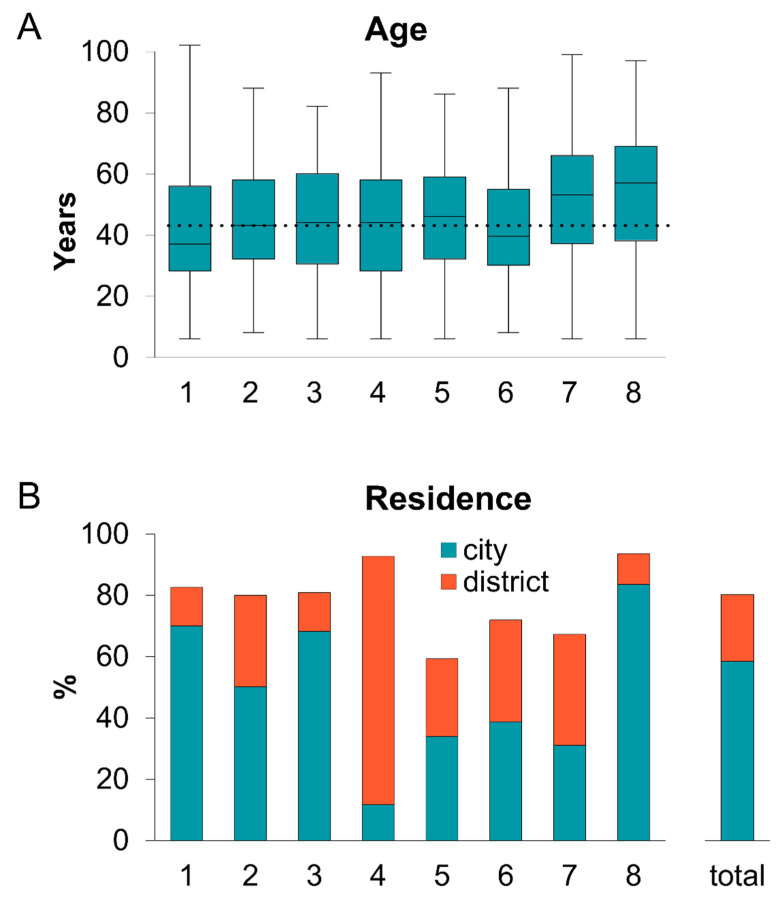
Age (**A**) and residence (**B**) of tested people at different locations. (**A**) Median age (with quartiles and whiskers) of people tested at all individual test centers; the dotted line indicates the overall median age. (**B**) Ratio between tested people originating from the city (turquoise) and district (orange) in the individual test centers and in total. 1: Central bus station, 2: office block, 3: theatre, 4: district community center, 5: park and ride, 6: suburban test center, 7: hospital, 8: bus.

**Figure 4 ijerph-19-04721-f004:**
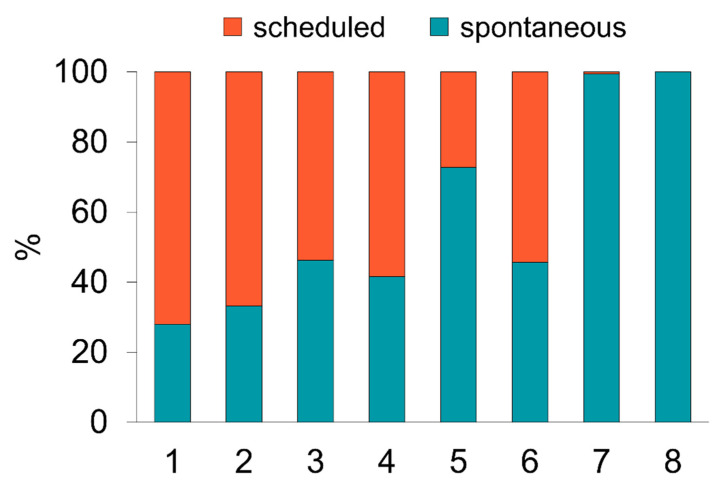
Scheduled versus spontaneous SARS-CoV-2 PoC antigen tests at different locations. Ratios of people tested with scheduled (orange) and spontaneous (turquoise) appointments in the individual test centers. 1: Central bus station, 2: office block, 3: theatre, 4: district community center, 5: park and ride, 6: suburban test center, 7: hospital, 8: bus.

**Figure 5 ijerph-19-04721-f005:**
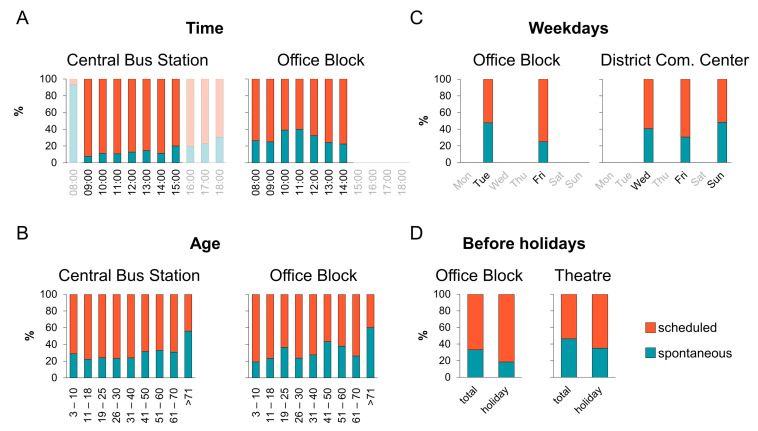
Scheduled versus spontaneous SARS-CoV-2 PoC antigen tests in relation to time, age, weekdays, and holidays. Selected ratios of tested people with scheduled (orange) and spontaneous appointment (turquoise) tests, analyzed for opening hours (**A**), age (**B**), weekdays (**C**), and before holidays (**D**). More details in Appendix A.

**Figure 6 ijerph-19-04721-f006:**
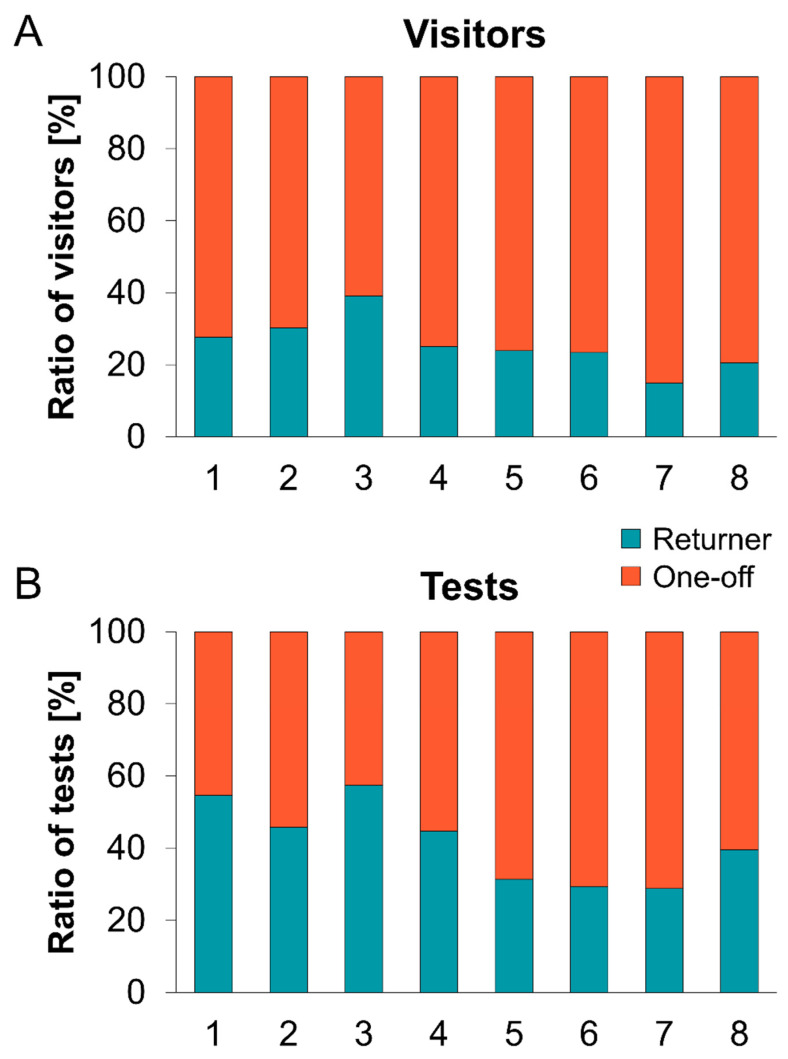
Ratios of returning to one-off visitors and respective SARS-CoV-2 PoC antigen tests at the different locations. (**A**) Ratios of visitors attending at least twice (returner, turquoise) and only once (one-off, orange) for all individual test centers. (**B**) Ratios of numbers of tests used by returning (turquoise) and one-off (orange) visitors for all individual test centers. 1: Central bus station, 2: office block, 3: theatre, 4: district community center, 5: park and ride, 6: suburban test center, 7: hospital, 8: bus.

**Table 1 ijerph-19-04721-t001:** Overview of tests centers regarding business hours and testing.

		Open Days	Opening Hours	Appointments	Number of Tests	Average Tests per Open Week	Average Tests per Open Day
**1**	**Central Bus Station**	Mon–Sat *	9.00–15.00 **	+	15,981	1332	219
**2**	**Office block**	Tue, Fri	9:00–15:00	+	1016	127	64
**3**	**Theater**	Variable	Variable	+	742	106	49
**4**	**District Community Center**	Wed, Fri, Sun	Wed: 16.00–19.00 Fri: 14.00–17.00 Sun: 12.00–15.00	+	1422	203	71
**5**	**Park and Ride**	Tue, Wed	9.00–15.00	+	577	96	48
**6**	**Suburban Test Center**	Wed, Sat	9.00–15.00	+	591	99	49
**7**	**Hospital**	Mon–Fri	7.00–15.00 ***	(-)	5116	731	171
**8**	**Bus**	Tue–Sat ****	9:30–11.00, 12:30–14:00	-	1924	192	43

* One Sunday in addition (Easter); ** four Fridays until 18:00; *** until 16:30 from 11th June; **** Saturday starting with calendar week 18; + scheduled appointments with booking software possible; - no scheduled appointments possible; (-) scheduled appointments with booking software possible, but information not released.

**Table 2 ijerph-19-04721-t002:** Open period of test centers.

		Calendar Weeks													
		13	14	15	16	17	18	19	20	21	22	23	24	25	26
		Date (begin)													
		29 March	5 April	12 April	19 April	26 April	2 May	10 May	17 May	24 May	31 May	7 June	14 June	21 June	28 June
**1**	**Central Bus Station**	+	+	+	+	+	+	+	+	+	+	+	+	+	+
**2**	**Office block**	-	-	-	-	+	+	+	+	+	+	+	+	+	+
**3**	**Theater**	-	-	-	-	-	+	+	+	+	+	+	+	-	-
**4**	**District Community Center**	-	-	-	-	-	+	+	+	+	+	+	+	+	+
**5**	**Park and Ride**	-	-	-	-	-	-	+	+	+	+	+	+	+	-
**6**	**Suburban Test Center**	-	-	-	-	-	-	+	+	+	+	+	+	+	-
**7**	**Hospital**	-	-	-	-	-	+	+	+	+	+	+	+	+	+
**8**	**Bus**	-	-	+	+	+	+	+	+	+	+	+	+	-	-

Analysis from calendar weeks 13 to 24; the period afterwards shows when test centers opened (+, turquoise), and whether locations were closed (-) or remained open after the analysis period (light blue).

**Table 3 ijerph-19-04721-t003:** Overlapping usage of test locations.

Test Centers (Number of Visitors)	1 (10,032)	2 (791)	3 (520)	4 (1055)	5 (523)	6 (546)	7 (4279)	8 (1469)
**1** **(10,032)**		**18%** **(145)**	**30%** **(156)**	8% (88)	17% (90)	14% (76)	2% (86)	7% (99)
**2** **(791)**	**1%** **(145)**		5% (24)	1% (15)	4% (21)	4% (20)	0.2% (8)	1% (13)
**3** **(520)**	**2%** **(156)**	3% (24)		1% (12)	4% (22)	2% (13)	**0.02%** **(1)**	0.5% (7)
**4** **(1055)**	1% (88)	2% (15)	2% (12)		2% (10)	4% (22)	0.2% (10)	1% (8)
**5** **(523)**	1% (90)	3% (21)	4% (22)	1% (10)		2% (9)	0.1% (6)	1% (8)
**6** **(546)**	1% (76)	3% (20)	3% (13)	2% (22)	2% (9)		**0.1%** **(4)**	0.5% (7)
**7** **(4279)**	1% (86)	1% (8)	**0.2%** **(1)**	1% (10)	1% (6)	**1%** **(4)**		1% (11)
**8** **(1469)**	1% (99)	2% (13)	1% (7)	1% (8)	2% (8)	1% (7)	0.3% (11)	

Numbers of absolute matches (indicated in brackets) relative to numbers of visitors per test location (indicated in brackets in the heading). For all combinations, two ratios are indicated in turquoise and orange for both locations. Examples mentioned in the text are highlighted in bold. Reference for calculation was the number of tests of the location in the first row (heading, light blue); numbers were rounded to zero decimal places (exceptions were values below 0.5%, which were rounded to one or two decimal places to distinguish them from ‘0’). 1: Central bus station, 2: office block, 3: theatre, 4: district community center, 5: park and ride, 6: suburban test center, 7: hospital, 8: bus.

## Data Availability

The dataset used during the current study is available from the corresponding author upon reasonable request.

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
