# Peer review of "Free PoC Testing for SARS-CoV-2 in Germany: Factors Expanding Access to Various Communities in a Medium-Sized City"

_ijerph, 2022, doi:10.3390/ijerph19084721_

Round 1
Reviewer 1 Report
Antigen testing is a crucial step towards the fight of Covid-19. The majority of manuscripts are currently focused on vaccination rather than on testing. The current manuscript covers an interesting area and investigates a range of parameters related to antigen testing in Germany. Overall the manuscript is well written, covers in detail the investigation and includes statistics well supported by graphs. However, the importance of antigen testing (or generally the testing) in COVID-19 is poorly described through the manuscript (specifically in introduction and discussion). Thus, I am recommending more research to cover the particular parameter and improve the flow of manuscript. Please see below my comments in detail.
Introduction:The importance of antigen testing during Covid-19 pandemic is missing from the introduction. Why is important? Why countries and particularly Germany should invest on free testing? How it might reduce the cases? Briefly the advantages and disadvantages of antigen testing? Can antigen testing be as important as vaccination? Any previous studies on testing should be mentioned (nationally or internationally).
Materials and Methods: A range of parameters such as age, locations, time period and region are well described. Please refer in detail, why the particular locations have been selected and how are related to the current study.
Discussion: The relationship between free testing and COVID-19 reported cases is missing from the manuscript. Does the free testing caused a reduction on the daily cases? or affected a specific group of individuals. What was the overall impact?
Author Response
Introduction:The importance of antigen testing during Covid-19 pandemic is missing from the introduction. Why is important? (lines 38-65) Why countries and particularly Germany should invest on free testing? (lines 59-63) How it might reduce the cases? (lines 42-46) Briefly the advantages and disadvantages of antigen testing? (lines 46-51 and 54-63) Can antigen testing be as important as vaccination? (line 41-46) Any previous studies on testing should be mentioned (nationally or internationally). Thank you for the helpful comment, we improved the section regarding the mentioned topics and added the context to previous studies on community-based testing.
Materials and Methods: A range of parameters such as age, locations, time period and region are well described. Please refer in detail, why the particular locations have been selected and how are related to the current study. We added the information in the Material and Method section (lines 99/100).
Discussion: The relationship between free testing and COVID-19 reported cases is missing from the manuscript. Does the free testing caused a reduction on the daily cases? or affected a specific group of individuals. What was the overall impact? We agree that showing a correlation between testing and the reduction of cases is a most relevant aspect. However, as mentioned in the discussion (lines 449-452) it was not possible with the used dataset. We hope that we can conclude such a relationship with bigger data sets over longer periods in future. But chances to unravel the precise contribution of testing even using extended datasets are limited, since so many measures and events are performed in parallel (lockdowns, testing, new variants, vaccination,…). We added this point to the introduction (line 63-65) and in an additional section of the limitations of the study (lines 480-486) in comparison with previous studies on testing.
Reviewer 2 Report
This study assesses factors that expand access to free Point of Care (PoC) Testing of SARS-CoV-2 to various communities in a medium-sized city in Germany. The paper aims to evaluate information on the use of test centers and individual factors of users to provide perspective to the cost and benefit aspect of this public health measure. The review is as follows:
- Lines 11 – 12 – Check wording in” Point of Care (PoC) antigen tests were offered to citizen at least once a week…”. The word should be plural (citizens).
- Lines 17-19 – The latter part of this sentence is unclear – “The centers were visited by different groups: some centers were preferred by a predominantly younger demographic, others by older age groups, such as a bus which reached districts with only few other test possibilities..” The statement ‘others by older age groups, such as a bus’ is unclear.
- Lines 34-35 – In “So far, more than five million deaths have been reported in association with COVID-19”, it needs to be contextualized and specified that the five million deaths are worldwide.
- The background and significance of this topic is not well established in the paper. Authors need to more clearly assert a need for this study and how this study adds to or addresses a gap in the existing literature.
- In the Introduction, authors should also present the COVID-19 numbers for Germany to provide context.
- The subheadings under Materials and Methods are not well-positioned or clearly labeled.
- For Materials and Methods, how were participants recruited?
- Lines 86-87 – In “All people agreed to the statistical analysis of their data with a privacy statement”, how was this agreement captured? Were forms signed?
- Line 133 – Check for spacing within the words ‘Suburban Test Center’.
- In Table 1 and for Appointments, what does the minus symbols [‘(-)’ and ‘-‘] denote? Does this mean appointments are not required?
Overall, this is a pertinent paper on a relevant topic. The Introduction and Methods need further development. Check organization of information in the Materials and Methods section and Table 1 to aid readability. The rationale for the study should be more compelling. Tending to these areas may help to improve the paper.
Author Response
- Lines 11 – 12 – Check wording in” Point of Care (PoC) antigen tests were offered to citizen at least once a week…”. The word should be plural (citizens). Thanks for the comment, we changed as suggested.
- Lines 17-19 – The latter part of this sentence is unclear – “The centers were visited by different groups: some centers were preferred by a predominantly younger demographic, others by older age groups, such as a bus which reached districts with only few other test possibilities..” The statement ‘others by older age groups, such as a bus’ is unclear. The statement is clarified.
- Lines 34-35 – In “So far, more than five million deaths have been reported in association with COVID-19”, it needs to be contextualized and specified that the five million deaths are worldwide. Thank you for this comment, we added the missing information.
- The background and significance of this topic is not well established in the paper. Authors need to more clearly assert a need for this study and how this study adds to or addresses a gap in the existing literature. We have extended the introduction and discussion to address this important comment.
- In the Introduction, authors should also present the COVID-19 numbers for Germany to provide context. Done as suggested in lines 35 -37.
- The subheadings under Materials and Methods are not well-positioned or clearly labeled. Thank you for the comment. We structured the paragraph as suggested
- For Materials and Methods, how were participants recruited? All data of tested people were recorded. We added the information in the material and methods section to clarify this issue (lines 110-112).
- Lines 86-87 – In “All people agreed to the statistical analysis of their data with a privacy statement”, how was this agreement captured? Were forms signed? With adding the information for the previous comment this is clarified.
- Line 133 – Check for spacing within the words ‘Suburban Test Center’ Done as suggested.
- In Table 1 and for Appointments, what does the minus symbols [‘(-)’ and ‘-‘] denote? Does this mean appointments are not required? Thank you for the comment, we added the information in table 1.
Overall, this is a pertinent paper on a relevant topic. The Introduction and Methods need further development. Check organization of information in the Materials and Methods section and Table 1 to aid readability. The rationale for the study should be more compelling. Tending to these areas may help to improve the paper. Appreciating the detailed and helpful comments, we improved all suggested sections and embedded this study more clearly into the context of existing studies (lines 39-65) to emphasize the rational (in particular lines 86-90).
Reviewer 3 Report
Manuscript ID: ijerph-1651843 seems an interesting research investigating 27369 datasets (corresponding to SARS-CoV-2 free antigen tests performed in a medium-seized European City) considering groups, timing, frequency, location of test centers, etc. Results could be valuable but in this form the manuscript is hard to follow. Let me highlight some aspects that may improve your manuscript:
- Please state appropriate number to paragraphs under section Materials and methods (Locations, Time period, Pseudonymized data, Analysis, Age, Region, Scheduled versus spontaneous tests, People and Metadata)
- Rows 108-112. There are presented some postal codes corresponding to urban district and rural district. Is this really necessary since no correlation with these codes is presented?
- According to Table 2 was used as mobile test center for 10 weeks. In figure 1 there are presented 9 different locations of the bus. The bus was moved weekly or not. You should detail this aspect.
- Figure 2 is very loaded. You should find a way to present data in a more clear way. Also local measures taken by authorities (rows 197-205) should be detailed in order to better interpretation of figure 2.
- You chose to present Figures 3-6, Tables 2 and 3 and all figures included in supplementary Materials in light grey and dark grey. I consider that you should use colors in order to highlight presented data/information.
- In figures S1-S4 and S6 data corresponding to 7. Hospital are not presented.
- Rows 412-416, you mention a parallel study, designed and managed by your research team, which is under review to another journal. This is inappropriate to cite this kind of findings in such a way.
- Even if along the article some limitations are mentioned I consider that it is necessary to built up a separate paragraph dedicated to limitations.
Author Response
- Please state appropriate number to paragraphs under section Materials and methods (Locations, Time period, Pseudonymized data, Analysis, Age, Region, Scheduled versus spontaneous tests, People and Metadata) Thank you for the comment. We structured the paragraph as suggested.
- Rows 108-112. There are presented some postal codes corresponding to urban district and rural district. Is this really necessary since no correlation with these codes is presented? We agree the information is redundant since the information which postal code is city and which greater region is public available. Therefore, we shortened this paragraph (lines 134-136).
- According to Table 2 was used as mobile test center for 10 weeks. In figure 1 there are presented 9 different locations of the bus. The bus was moved weekly or not. You should detail this aspect. We apologize not including the information of the bus’s locations. The Bus operated each week at 9 different locations. We added this information in the text (lines 184/185) and the legend of figure 1.
- Figure 2 is very loaded. You should find a way to present data in a more clear way. Also local measures taken by authorities (rows 197-205) should be detailed in order to better interpretation of figure 2. We agree there is much information in figure 2 demonstrating that many things happened in parallel. We now simplified and restructured the figure and rephrased the legend for the measures to better interpretation of the figure.
- You chose to present Figures 3-6, Tables 2 and 3 and all figures included in supplementary Materials in light grey and dark grey. I consider that you should use colors in order to highlight presented data/information. Thank you for this comment, we changed all figures to colors for highlighting those data as suggested. We realized by changing colors in the tables that we forgot to copy the table legends to the template. We completed them and apologize for the mishape!
- In figures S1-S4 and S6 data corresponding to 7. Hospital are not presented. Only 1 % (32 data sets) of the tests came scheduled to the hospitals test center as this option wasn´t communicated to the public. Therefore, we did not include this analysis since those data are not substantial.
- Rows 412-416, you mention a parallel study, designed and managed by your research team, which is under review to another journal. This is inappropriate to cite this kind of findings in such a way. Thank you for this notice. We rephrased this part and included the proper citation in the text.
- Even if along the article some limitations are mentioned I consider that it is necessary to built up a separate paragraph dedicated to limitations. We added a paragraph (lines 471-485) before the conclusion as suggested.
Round 2
Reviewer 1 Report
Dear authors
All the comments have been addressed and the manuscript has been improved. My major concern regarding the importance of antigen testing in introduction, now is well covered and provides a clear background information to the readers.
Author Response
Thank you for your comment. We appreciate that we could considerably improve our manuscript with the help of your feedback.
Reviewer 2 Report
The authors have done well to incorporate the suggested feedback. The revised manuscript is clearer. After a proofread of the manuscript, it will appear suitable for publication.
Author Response

(The authors gave the same response as above.)

Reviewer 3 Report
Dear authors,
You carefully answered my questions and requests. The manuscript was very much improved and I would like to recommend its publication after a minor revision.
Please update all explanatory texts related to figures (manuscript and supplementary materials) since at this moment term like "dark grey" and "light grey" are used (rows 272, 314, 322, 359-361 of the manuscript version 2.0).
Also figure S6 is somehow upside down. Please be kind and correct this.
Author Response
Thank you for your comments! We changed the mentioned parts and apologize that we missed to change it immediately. Thanks to your reviews, we could considerably improve our manuscript!